# Influence of a Polyherbal Choline Source in Dogs: Body Weight Changes, Blood Metabolites, and Gene Expression

**DOI:** 10.3390/ani12101313

**Published:** 2022-05-20

**Authors:** Germán David Mendoza-Martínez, Pedro Abel Hernández-García, Fernando Xicoténcatl Plata-Pérez, José Antonio Martínez-García, Augusto Cesar Lizarazo-Chaparro, Ismael Martínez-Cortes, Marcia Campillo-Navarro, Héctor Aarón Lee-Rangel, María Eugenia De la Torre-Hernández, Adrian Gloria-Trujillo

**Affiliations:** 1Departamento de Producción Agrícola y Animal, Universidad Autónoma Metropolitana—Xochimilco, Mexico City 04960, Mexico; gmendoza@correo.xoc.uam.mx (G.D.M.-M.); ppfx2221@correo.xoc.uam.mx (F.X.P.-P.); jamgar@correo.xoc.uam.mx (J.A.M.-G.); ismaelmartinezcortes@gmail.com (I.M.-C.); mdelatorre@correo.xoc.uam.mx (M.E.D.l.T.-H.); 2Centro Universitario Amecameca, Universidad Autónoma del Estado de México, Amecameca 56900, Mexico; pedro_abel@yahoo.com; 3Centro de Enseñanza, Investigación y Extensión en Producción y Salud Animal (CEPIPSA), Universidad Nacional Autónoma de México, Tlalpan, Mexico City 14500, Mexico; lizarazo@unam.mx; 4Coordinación de Investigación del Centro Médico Nacional 20 de Noviembre, ISSSTE, Mexico City 03100, Mexico; marcy265@hotmail.com; 5Facultad de Agronomía y Veterinaria, Universidad Autónoma de San Luis Potosí, Carretera Federal 57 Km 14.5, Ejido Palma de la Cruz, Soledad de Graciano Sánchez, San Luis Potosi City 78321, Mexico; leehec@hotmail.com; 6Consejo Nacional de Ciencia y Tecnología–Universidad Autónoma Metropolitana-Xochimilco, Mexico City 03940, Mexico

**Keywords:** *Canis familiaris*, health, nutrigenomics, feed plant additive

## Abstract

**Simple Summary:**

In pet foods, choline is supplied as choline chloride. This salt is hygroscopic and makes food processing difficult. This has led to its replacement, in livestock farms, by chemically stable herbal additives rich in phosphatidylcholine. The herbal composition and the contribution of secondary metabolites give the additive nutraceutical properties that could benefit the animal’s health. In this sense, increased levels of a polyherbal additive were supplied in dog diets, and their effects were compared to diets with choline chloride and without extra choline. The results showed that the animal response was the same with the two sources of choline. However, the herbal additive at the gene level demonstrated properties to prevent cardiovascular and metabolic diseases, cancer prevention, inflammatory and immune response, and behavior and cognitive processes in dogs.

**Abstract:**

Choline chloride is used to provide choline in dog foods; however, in other domestic species, it has been replaced with a polyherbal containing phosphatidylcholine. A polyherbal containing *Achyrantes aspera*, *Trachyspermum ammi*, *Citrullus colocynthis*, *Andrographis paniculata*, and *Azadirachta indica* was evaluated in adult dogs through body weight changes, subcutaneous fat thickness, blood metabolites, and gene expression. Forty dogs (4.6 ± 1.6 years old) who were individually housed in concrete kennels were randomly assigned to the following treatments: unsupplemented diet (377 mg choline/kg), choline chloride (3850 mg/kg equivalent to 2000 mg choline/kg diet), and polyherbal (200, 400, and 800 mg/kg) for 60 days. Blood samples were collected on day 59 for biochemistry, biometry, and gene expression analysis through microarray assays. Intake, final body weight, and weight changes were similar for the two choline sources. Feed intake variation among dogs (*p* = 0.01) and dorsal fat (*p* = 0.03) showed a quadratic response to herbal choline. Dogs that received the polyherbal diet had reduced blood cholesterol levels (Quadratic, *p* = 0.02). The gene ontology analysis indicated that 15 biological processes were modified (*p* ≤ 0.05) with implications for preventing cardiovascular and metabolic diseases, cancer prevention, inflammatory and immune response, and behavior and cognitive process. According to these results that were observed in a 60 day trial, the polyherbal form could replace choline chloride in dog diets at a concentration of 400 mg/kg.

## 1. Introduction

Choline chloride is commonly used as choline in dog food; however, its high hygroscopicity and acceleration of oxidative loss [1] have led to it being replaced in other domestic species with a standardized polyherbal mixture containing phosphatidylcholine (PtdCho) [2,3,4,5]. An experiment with dogs showed that 500 mg polyherbal/kg of diet replaced 2000 mg synthetic choline/kg of diet (contributed by choline chloride) without affecting the food preference of dogs [6]. Another experiment confirmed that 400 mg polyherbal/kg of diet could replace choline chloride in adult dog diets without affecting food preference, nutrient digestibility, and fecal characteristics [7]. The polyherbal has been characterized and, in addition to providing PtdCho, contains methylated metabolites [8] and volatile metabolites with nutraceutical properties [9], which improve growth and health status while modifying gene expression in calves [10].

Research in broilers and ruminants [2,3,4,5] has demonstrated that PtdCho can meet choline requirements. This opens the need to evaluate polyherbal supplements as a source of choline conjugate in companion animals. In dogs, the requirements of choline [11,12], according to the choline content in homemade diets [13] or results from experiments with obese dogs [14], could be overestimated. Foods differ in their concentrations of free choline, glycerophosphocholine, phosphocholine, PtdCho, and sphingomyelin [15], and these forms vary in their bioavailability [16,17] and metabolism [1,18].

The understanding of the PtdCho as a key intermediate metabolite [19] raises the need to evaluate its different forms in order to meet the physiological choline needs in the organism (acetylcholine, phosphatidylcholine, sphingomyelin, lisophosphatidylcholine dipalmitoylphosphatidylcholine, betaine, platelet-activating factor, etc.). PtdCho can be incorporated into the cell’s metabolism and save energy by not being supplemented in its precursor forms (choline, acetylcholine, betaine, sphingomyelin) [20], thus meeting requirements for biological membrane functions and improving antioxidant status at the cellular level [21].

Therefore, the objective of this experiment was to evaluate the metabolism and health of dogs through measurements of blood chemistry, blood biometry, liver ultrasound, gene expression, and body weight changes, in dogs, supplemented with increasing dietary levels of a polyherbal compound (*Achyranthes aspera*, *Azadirachta indica*, *Andrographis paniculata*, *Citrullus colocynthis*, and *Trachyspermum ammi*) versus a diet with choline chloride or an unsupplemented diet.

## 2. Materials and Methods

### 2.1. Study Design, Animals, and Treatments

The experiment had a duration of 60 days and was conducted in the Centro de Investigación en Alimentos para Mascotas (CIAM), in Tepeji del Rio, Hidalgo, México (19°54′14′′ N, 99°20′29′′ W; elevation 2175 m). The average temperature is 15.2 °C (3 °C min–30 °C max), average humidity is 5% (0% min–10% max), and has light:dark cycles of 10:13 h.

Forty (twenty males and twenty females) healthy adult dogs (*Canis lupus familiaris*; 4.6 ± 1.6 years old) were individually housed in concrete kennels (2 m width × 5 m long). The breeds used were: Beagles, Schnauzer, Bichon Frize, Dachshund, Airedale Terrier, and Jack Russell. Dogs were fed individually and were randomly allocated to the following dietary treatments: (1) unsupplemented diet (377 mg choline naturally derived from the diet/kg of food), (2) choline chloride (3850 mg synthetic product equivalents 2000 mg of choline/kg of food), and (3) three levels of polyherbal supplement (BioCholine; 200, 400, and 800 mg/kg) with a contribution of 1.6% phosphatidylcholine.

The basal diet was a complete dry extruded dog food for maintenance, prepared as adult kibble (Dosky PT290, Canis Food Nutrition) that was formulated in accordance with the nutritional recommendations of NRC [11] and the Association of American Feed Control Officials [12] for adult dogs. The ingredients and chemical composition of the basal diet are reported in Mendoza et al. [7]. The vitamin and mineral premixes were manufactured by DSM Nutritional Products of Mexico, where polyherbal (Nuproxa México–Switzerland) and choline chloride (60% choline, DSM) were used to prepare the dietary treatments. Diets were stored for 15 days in sealed bags, and were protected from direct sunlight at a room temperature of 20.5 ± 1.2 °C and a relative humidity of 68.5 ± 2.3% before being supplied to the dogs.

Daily food intake was recorded, and consumption variation between days was evaluated as an indicator of stability and animal welfare [10]. Animals were fed once per day (10:00 am every day) in quantities to meet their metabolizable energy requirement (MER, Kcal/d = 130 × (Body weight, kg)^0.75^) as recommended by NRC [11], adjusting for activity given their 40 min of daily exercise. Clean water was provided ad libitum.

### 2.2. Ultrasound Study

On day 50 of the experiment, to measure backfat and to evaluate images of liver integrity, a portable ultrasound scanner (Chison Eco 6^®^; Chison Medical Technologies Co., Jiangsu, China) equipped with a multi-frequency linear transducer (5.3 MHz–10.0 MHz Linear L7M-A; Chison Medical Technologies Co., Jiangsu, China) was used. Dogs were prepared by cutting their hair prior to evaluation. To measure backfat, the equipment was used in the lumbar region because it has been reported to have less variation, and the results have been correlated with the body condition of dogs [22]. Liver ultrasounds were used to search for the presence of normal or abnormal characteristics according to Kemp et al. [23], e.g., abnormal echogenicity, masses, nodules, mottled parenchyma, or normal findings.

### 2.3. Biochemical and Hematological Profiles

On day 59 of the experiment, pre-prandial blood samples of all dogs were collected from the jugular vein using vacutainer tubes with sodium citrate, EDTA, and without anticoagulant. The blood samples were transported under refrigeration (4 °C) to the veterinary laboratory Laclivet for immediate processing. Tubes without anticoagulant were centrifuged in a refrigerated centrifuge (Sigma 2–16 k, Sigma Laborzentrifugen GmbH, Osterode am Harz, Germany) at 3500× *g* for 20 min to obtain blood serum, which was stored in conical tubes and kept in a freezer at −20 °C until analysis. Cholesterol, glucose, total protein, albumin, bilirubin, alkaline phosphatase (ALP), lactate dehydrogenase (LDH), aspartate amino transferase (AST), calcium, and phosphorus were determined with an autoanalyzer Kontrolab^®^ (Hamburg, Germany) (2017). The blood sample collected in disodium–EDTA was used for hematocrit and blood count with a hematology analyzer EasyVet^®^ (QS Kontrolab, Hamburg, Germany).

### 2.4. Blood Samples, RNA Extraction, and Microarray

From dogs supplemented with 2000 mg synthetic choline and 800 mg polyherbal, RNA was extracted from whole blood following the protocol described by Winn et al. [24]. By triplicate, 350 µL of blood was transferred immediately to a 2 mL microtube (O-ring cap) with 1 mL of Trizol (Invitrogen, Waltham, MA, USA). Samples were gently homogenized by inverting the tubes and placing them on ice for transport. In the laboratory, total RNA was extracted following the protocol that was proposed by the manufacturer (Invitrogen, Waltham, MA, USA). The integrity, purity, and concentration of the RNA were evaluated as described by Díaz et al. [10].

For transcriptome analysis by DNA microarray, two pools of 30,000 ng of RNA were obtained. Each pool was made up of equal concentrations of RNA (3750 ng RNA) from each repetition (8 animals) that made up the experimental group (800 mg polyherbal vs. 3850 mg choline chloride) to be evaluated. The transcriptome analysis was carried out in the Microarray Unit of the Institute of Cellular Physiology of the National Autonomous University of Mexico (UNAM), where a heterologous mouse chip (M22K) was used with 24,341 applications. The results of the microarray were analyzed with the GenArise software to identify differentially expressed genes (DEG; z-score ≥ 1.5). For the gene set enrichment, the DAVID 6.8 bioinformatics tool (Database for Annotation, Visualization, and Integrated Discovery) [25] was used. Gene set enrichment analysis using gene ontology was used to summarize biological meaning from the identified differentially expressed transcripts.

### 2.5. Statistical Analyses

#### 2.5.1. Animal Performance, Energy Balance, and Blood Variables

The experiment was a completely random design with eight repetitions per treatment. A Shapiro–Wilk test was used to test the normal distribution of variables. Data were analyzed using R software (v. 2.15.3; 2013) [26], and orthogonal linear and quadratic polynomials were used to evaluate the effects of the polyherbal additive. The model used was: Yij = µ + τ i + eij, in which µ is the mean value, τ i is the treatment effect (fixed), and eij is the error term. A statistically significant effect was considered with a probability of ≤0.05, while a trend was considered with a probability of between 0.051 and 0.1.

#### 2.5.2. Gene Enrichment Analysis

For gene enrichment analysis, a minimum of 2 genes per biological group was considered. An EASE score (modified Fisher Exact) with a probability of ≤0.05 and the fold enrichment value were considered to identify the enriched annotation terms that belonged to the evaluated gene groups and biological closeness between the enriched processes [25], respectively.

## 3. Results

### 3.1. Intake, Liveweight Changes, and Dorsal Fat

No changes were observed (*p* = 0.75) in feed intake due to the type of choline supplement (Table 1). However, the inclusion of 400 mg of polyherbal/kg of feed showed (Linear, *p* < 0.02) the lowest variation in consumption between days within each animal (Table 1). No treatment differences were observed in daily weight changes (*p* = 0.20), average daily gain (*p* = 0.62), or energy expenditure (*p* = 0.70). The polyherbal supplement showed a quadratic effect (*p* = 0.03, Table 1) on back fat, which was greater when compared to choline chloride (contrast, *p* = 0.05).

### 3.2. Biochemical, Hematological Profiles, and Ultrasound

Dogs that were fed diets with choline sources presented lower cholesterol levels than the control (quadratic response; *p* = 0.02; Table 2). In other parameters, minor differences were detected, where the polyherbal tended to induce a greater concentration of globulins (1.94%) than choline chloride (*p* = 0.08, Table 2). No treatment differences (*p* > 0.05) were observed in the liver enzymes ALP, LDH, and AST (Table 2).

The hematological studies of the dogs showed that the polyherbal tended to increase the percentage of eosinophils when compared to choline chloride (*p* = 0.07, Table 3). No liver abnormalities (data not shown) were detected by an ultrasound of the liver, confirming that the dogs were healthy, which coincided with the hematological values.

### 3.3. Microarrays

Microarray results confirm that polyherbal can replace choline chloride without changes in 19,417 genes, representing about 90.5% of the canine genome. From 23,232 genes evaluated, the polyherbal affected the expression of 2207 genes (Appendix A) when compared to choline chloride. Considering the total assessed transcripts by expression level, 46.0 and 32.6% of the genes were under and overexpressed, respectively, at least two times versus choline chloride. The gene ontology analysis represented 15 enriched biological processes (*p* < 0.05; Table 4), which were grouped according to the possible implications for the dogs in four categories: (a) prevention of cardiovascular and metabolic diseases; (b) cancer prevention; (c) inflammatory and immune response; (d) behavior and cognitive process (Table 4).

The polyherbal, compared to choline chloride, reduced the expression of genes of the renin–angiotensin system (Fc: 3.4, *p* = 0.03; Table 5), the digestion and absorption of carbohydrates (Fc: 3.3, *p* = 0.02; Table 6), and the peroxisome proliferator-activated receptor signaling pathways (PPAR, Fc: 2.5, *p* = 0.02, Table 5), implied in the group of cardiovascular and metabolic diseases prevention (Table 4).

The polyherbal affected genes that were related to cancer prevention (Fc: 1.5, *p* = 0.04, Table 6), where eight biological processes in cancer pathways were involved (regulation of stem cell pluripotency, transforming growth factor β, forkhead box O, mitogen-activated protein kinase, transcriptional dysregulation in cancer, estrogens, cholinergic synapses). In the genes related to immune and inflammatory response (Table 7), the polyherbal, compared to choline chloride, showed a decrease in the chemokine signaling pathway and some underexpression of genes in the adhesion junctions process (Fc: 2.5, *p* = 0.03).

The polyherbal reduced the expression of GABA (γ-aminobutyric acid) genes grouped in the biological process called nicotine addiction (Fc: 3.5, *p* = 0.01, Table 8), which are the genes involved in behavior and cognitive processes.

## 4. Discussion

### 4.1. Intake, Live Weight Changes, and Dorsal Fat

The polyherbal did not affect intake as previously reported [7]. The final weight shows that the dogs were fed around the maintenance requirement. At the intermediate concentrations of the polyherbal (200 and 400 mg/kg), the changes in body condition estimated with the back fat show that choline chloride can be substituted. As a choline replacement, the polyherbal provides phosphatidylcholine, which follows a different metabolic pathway than free choline, expending less metabolic energy, which is thus available to the cells. Free choline requires ATP-dependent transporters and an additional ATP molecule to form phosphocholine [19]. At the same time, phosphatidylcholine is absorbed with other products of fat digestion, is transported in the blood as lipoproteins, and is available to cells and tissues directly [27].

Among the metabolites of the polyherbal, hexadecanoic acid (C16:0) [8] stands out, which can act as a methyl donor [28] and can save methionine. In other species, choline has been substituted with the same polyherbal [3,4,29], and the best growth was associated with the overexpression of the *PEPCK* gene, which affects the use of amino acids and glucose, resulting in higher energy and amino acids for growth and protein synthesis [30], respectively.

The polyherbal in intermediate doses improved body condition through the consideration of backfat as an indicator, which is within the values reported as healthy and below those of obese dogs [22]. The lipotropic effects of choline or phosphatidylcholine should be evaluated as an alternative to prevent obesity, a common nutritional disorder in dogs that affects health, well-being [31,32], longevity [33], and a higher incidence of metabolic diseases, hypertension, and neoplasms [31,33].

### 4.2. Biochemical, Hematological Profiles, and Ultrasound

The effects of choline chloride or polyherbal on cholesterol may be due to the functions of phosphatidylcholine, which participates in the synthesis and export of triglycerides in very-low-density lipoproteins [34], and this can affect plasma cholesterol, as reported in choline-supplemented dairy cattle [35].

In general, the blood count changes showed no immune response activation, and the results were within normal values [36]. The NRC of Dogs [11] compiles information from studies from the 1940s, where high doses of soybean phosphatidylcholine caused a decrease in circulating erythrocytes because the biological equivalence of phosphatidylcholine was not considered greater than choline chloride.

An ultrasound indicated the absence of liver problems, a result confirmed by biochemical (AST, ALT, and bilirubin) and hematological data [37,38]. No problems were detected in the control group, although the choline intake was below the requirements recommended by the NRC.

### 4.3. Microarrays

#### 4.3.1. Prevention of Cardiovascular and Metabolic Diseases

Polyherbal reduced the expression of genes that were involved in the renin–angiotensin system, whose high activity leads to vasoconstriction and sodium and water retention [39]. This polyherbal effect could lower the risk of cardiovascular disease and renal or cerebrovascular failure in dogs [40]—common conditions in veterinary medicine [41].

Another common condition in dogs is the incidence of Type II diabetes mellitus [42]. The decreased expression of the *G6PC* and *G6PC2* genes, which are involved in the digestion and absorption of carbohydrates (with the inclusion of polyherbal), reduces the presence of abnormalities associated with Type II diabetes mellitus. This has been observed with the overexpression of *G6PASE* on hepatocytes under in vitro conditions [43]. Similarly, the polyherbal reduced the expression of genes that code for glucose transporter proteins, such as the *SLC37A4* gene [44] and the *SLC2A2* gene [45]. This confirms the hypoglycemic capacity of the polyherbal that has been observed in weaning calves [10].

Polyherbal affected the expression of genes involved in the PPAR signaling pathway, which is important for its cholesterol modulating role and its implications in dyslipidemia [46,47]. The *PLTP* (phospholipid transfer protein) gene mediates the transfer of phospholipids and free cholesterol from triglyceride-rich lipoproteins into HDL. Although underexpression of *PLTP* has not yielded conclusive results in animal models, other functions have been recognized in vascular compartment lipid transport, impacting diseases such as atherosclerosis, cancer, and neurodegenerative disease [48]. The *CYP8B1* gene codes for the cytochrome P450 protein, whose function is to catalyze the conversion of steroids determining the relative amount of bile acids (cholic acid and chenodeoxycholic acid); bile acids affect the solubility of cholesterol, promoting its absorption. The Cyp8b1–P450 pathway was also downregulated with the polyherbal. The reduced expression of the *CYP8B1* gene has been of interest for pharmacological therapies since, at an experimental level, the incidence of fatty liver, lowered cholesterol absorption, prevention of atherosclerosis, hypercholesterolemia, and formation of gallstones in diabetic mice, as well as glucose homeostasis, have been reduced [46,49]. This could predict the presence of heart attacks [50].

The downregulation of *CYP4A10* with the polyherbal could be beneficial since it is overexpressed in hepatic steatosis and steatohepatitis conditions. In mouse studies, CYP4A fatty acid ω-hydroxylase P450s has been shown to play an essential role in the development of steatohepatitis [47]. However, the herbal composition of the additive and the secondary metabolites that compose it affect the transcriptional factor PPAR, in that they act as partial agonists and improve metabolic parameters in diabetic animal models [51].

The genes previously documented highlight that the polyherbal could reduce problems of metabolic syndrome, arterial hypertension, and obesity, as well as hypothetically impact the well-being and longevity of the dog.

#### 4.3.2. Cancer Prevention

The impact of polyherbal on cancer pathway genes shows a potential that should be evaluated in terms of its preventive effects on cancer problems with long-term studies. Genes that were underexpressed by the polyherbal included: hematopoietic and lymphoid tissues, such as acute myeloid leukemia (*IL3*, *RUNX1T1*, *CD14*, *BCL2A1*), B lymphoblastic lymphoma (*ETV6*, *IL3*, *ELANE*, *KMT2A*, *MLLT3*, *HMGA2*, *KDM6A*), T lymphoblastic lymphoma (*KMT2A*, *MLLT3*, *CDKN2C*, *HHEX*), Hodgkin lymphoma (*C-REL*, *BCL2A1*, *TRAF1*), and multiple myeloma (*NSD2*, *HIST1H3G*, *ITGB7*); epithelial cancers, such as prostate cancer (*PLAU*, *MMP3*, *MMP9*); neuroendocrine cancers, such as neuroblastoma (*PTK2*, *BMI1*), carcinoid (*KMT2A*), and Ewing’s sarcoma (*FLI1*, *ID2*, *TGFBR2*). Several herbs have been used to assist in cancer prevention in dogs [52], but studies are required to confirm anecdotal evidence from veterinarians using herbal complementary and alternative medicines among cancer patients.

The polyherbal is composed of, among other plants, *Andrographis paniculata* and *Azadirachta indica*. *Andrographis paniculata* exerts direct anticancer activity by arresting the cell cycle in the G0/G1 phase through the induction of the cell cycle inhibitory protein p27 and the decrease in the expression of cyclin-dependent kinase 4 (*CDK4*) [53], through the observed increase, with the polyherbal, of the cyclin-dependent kinase inhibitor 2C gene (*CDKN2C*, Fc: 2.7).

The *Azadirachta indica* stimulates apoptosis [54], which explains the effect of the polyherbal on the downward expression of the proto-oncogene viral thymoma 1 (*AKT1*) through the AKT and ERK pathways (extracellular signal-regulated kinases) [55]. Inhibition of cell apoptosis, together with abnormal cell proliferation, leads to cancer—the leading cause of death in companion animals [56,57].

The under-expression of the genes *NRAS*, *AKT1*, and *FKBP4*, involved in the estrogen signaling pathway, could prevent the development of mammary tumors in female dogs, since high levels of expression of *NRAS* and *AKT1* are related to carcinogenesis [58], while *FKBP4* is related to tumorigenesis, which is why it is common to observe them in hormone-dependent cancers, such as breast and prostate cancer [59]. Another potential benefit of the polyherbal in dogs would be its effect on signaling pathways that regulate stem cell pluripotency. The polyherbal reduced the expression of the *WNT1*, *WNT7B*, *APC*, and *AXIN* genes with respect to choline chloride. Aberrant activation of Wnt/b-catenin signaling leads to the progression of canine malignant melanoma [60], where 30% of metastases trigger b-catenin concentrations in the nucleus and cytoplasm [61]. However, the b-catenin gene (*CTNND1*) was downregulated by the polyherbal.

Polyherbal stimulated the expression of genes in the MAPK signaling pathway, which could be explained by some of the effects of its herbs, such as *Azadirachta indicate*. The MAPK pathway inhibits cell proliferation by the induction of protein kinases that are activated by AMP induction [62], as well as the induction of autophagy by the activation of the p38 MAPK pathway [54]. These are important cascades in regulating the proliferation, differentiation, apoptosis, and response to stress. It is difficult to predict the polyherbal’s impact on the MAPK/ERK pathway, since it has shown oncogenic and tumor suppressor effects depending on the specific tumor microenvironment [63].

Speculating on the effects of gene downregulation in the FoxO pathway is complicated. Still, it is recognized that FoxO transcription factors function as regulators of cellular homeostasis and are tumor suppressors [64]. It is also recognized that epigenetic dysregulation is one of the main causes of cancer [65]. The *BCL6* gene (understimulated by polyherbal) has been identified as a proto-oncogene [66], and its transcription is stimulated by *FOXO* genes. The expression of the *MMP3* gene, which is related to cytokines, tumor promoters, and oncogenic products, was also decreased [67]; the effect could be a reduction in the tumor-promoting function of FoxO.

The biological process of adhesion junctions was downregulated by the polyherbal. In this process, the underexpression of the *AFDN* gene stands out (related to the increase in cell motility and invasion [68]), as well as the *NECTIN2* gene (whose underexpression is indicative of the absence of cancerous tissues [69]), and the *SMAD2* gene (reported to affect TGFβ differently from *SMAD3* and *SAMD4* [70]).

The underexpression of genes in the TGFβ pathway could be of interest, since this pathway is involved in gastrointestinal stromal tumor interactions from early to late tumor stages, creating a favorable microenvironment for tumor initiation and cancer cell growth. Although anticancer roles are recognized at the cellular level depending on the tumor’s stage and genetic alteration, clinical studies show that TGFβ inhibition has the potential to control and delay some tumors [71]. The *SMAD7* gene was downregulated by the polyherbal, which could be positive, since its overexpression is common in many types of cancer, and its abundance is positively correlated with malignancy [72].

Therefore, all the data together indicate that the various genes that participate in cancer generation are affected by polyherbal consumption, especially those closely related to different types of cancer that occur more frequently in dogs and which cause high mortality [73]. Polyherbal is a possible cancer preventive treatment that could greatly benefit dog breeds with a higher risk of suffering from oncological diseases [74].

#### 4.3.3. Inflammatory and Immune Response

The immune response involves a wide variety of cells, adhesion molecules, chemokines, and cytokines to defend itself against aggression from the external environment, such as infections, by exerting an inflammatory response to eliminate pathogens and harmful molecules and culminating in its regulation to maintain body homeostasis [75].

Cytokines act as regulators of immune and inflammatory responses and intervene as growth factors of different cells, among which the hematopoietic ones stand out [76]. The downregulation of the *IFN*-γ gene is likely not positive, since this gene participates in diverse biological actions, including activating cells during the inflammatory process (monocytes, natural killer cells, basophils, eosinophils) with antiviral, immunomodulatory, and antiproliferative effects [77]. The gene-downregulating effect of the polyherbal may be of little impact since *IL-3*-deficient mice had no defect in hematopoiesis [78], even though *IL-3* contributes to leukocyte production, proliferation, and survival, and *IL-3* potentiates inflammation in sepsis has been shown [79].

The polyherbal reduced the expression of some chemokines, CXC (classified as inflammatory, homeostatic, and some with a dual function of being both inflammatory and homeostatic), and of some CC (chemotactic for monocytes, with some being plasmatic or platelet-related) [80]. The receptors of these functions are recognized in leukocyte recruitment and angiogenesis [81] as gene underexpression affects the known functions of specific genes (*CXCL9* and *CXCL11*: Th1 response, Th1, CD8, and NK trafficking; *CCL12*: monocyte trafficking; *CCL19*: homing to lymph node), being able to modify the immunoregulatory processes of these cytokines [82].

Among the genes that were downregulated with polyherbal is *AFDN2*, whose proangiogenic effects are of interest given that it contributes to physiological and pathological conditions such as atherosclerosis [83]. However, it is only a downregulation, given that the *AFDN* gene participates in other processes, such as the hemostatic function of the endothelial barrier [84].

Some *TNF* genes were downregulated by the polyherbal and participate in regulating immune cell functions and apoptosis [85]. The answer may not be favorable, since TNFs are inflammatory cytokines involved in suppressing infections [86].

Among the genes that were downregulated with the polyherbal and were related to the development of glaucoma are *BMP2* and *IL3*, identified by bioinformatics analysis [87].

Therefore, these findings indicate that the polyherbal exerts an anti-inflammatory effect and could contribute to the regulation of the immune response in different common inflammatory pathologies in dogs, such as inflammatory bowel disease.

#### 4.3.4. Behavior and Cognitive Process

The reduction of the expression of genes linked to the cholinergic synapse could have contrast effects. In the first place, it could have a beneficial effect on the stress levels of dogs. Stress impacts the GABAergic system and increases GABAergic transmission, elevates the baseline GABAergic response, and enhances the responsiveness of GABA receptors [88]. Secondly, downregulation of the *CHRNA7* gene could not be beneficial due to its neuroprotective, memory, and alertness functions [89,90,91,92]. It is probable that the underexpression of the two *SLC* genes may affect some central GABA neurons due to their specificity relating to membrane-bound transporters [93]. Slc17 is involved in storing the neurotransmitter glutamate and the metabolism of glycoproteins [94] when Slc32 transports amino acids across synaptic vesicles [95]. However, it is important to evaluate the effect of polyherbal supplementation on the behavior and cognitive process in dogs. A stress reduction in dogs could reduce its negative impacts reported on health, welfare, behavior, and lifespan [96].

### 4.4. Practical Implications for Feeding Dogs

The results of changes in weight, glucose, and blood cholesterol levels of the group without choline showed that it is required to incorporate phosphatidylcholine or choline chloride in the dog’s food and are similar to those of dogs with intakes below 40 mg/d [14]. The herbal product does not have the hygroscopic problems that choline chloride has [97], and polyherbal requires less space than choline chloride.

## 5. Conclusions

The polyherbal at 400 mg/kg of diet, evaluated with phosphatidylcholine and other secondary metabolites in a 60-day trial, showed that its use is safe because no toxic effects were observed. At this dose, it can be used to replace choline chloride in adult dogs. The potential advantages over choline chloride in health effects must be evaluated in long-term studies and other life stages.

## Figures and Tables

**Table 1 animals-12-01313-t001:** Effect of polyherbal level versus choline chloride (mg/kg) on feed intake (FI), body weight (BW) changes, dorsal fat, and energy intake.

	Polyherbal				Choline Chloride ^Δ^		*p*-Value		
Item	0	200	400	800	3850	SEM	L	Q	K *
FI g/d	239.4	235.5	236.2	229.4	238.9	20.51	0.75	0.95	0.82
FI variation among dogs %	10.00	10.40	7.14	18.28	9.10	2.13	0.02	0.01	0.25
Initial BW kg	10.70	10.24	10.28	10.06	10.56	1.297	0.85	0.70	0.68
Final BW kg	10.66	10.56	10.52	9.90	10.41	1.422	0.41	0.11	0.60
ADG g/d	−0.52	6.25	3.95	−2.81	0.93	6.175	0.62	0.20	0.85
Dorsal fat mm	2.77	3.14	3.22	2.86	2.64	0.283	0.66	0.03	0.05
Energy expenditure Kcal/d	900.4	862.4	873.0	847.7	893.9	85.59	0.70	0.94	0.69

* Contrast: Polyherbal vs. synthetic choline; SEM: Standard error of the mean; L: linear effect; Q: quadratic effect; FI: feed intake; ADG: average daily gain; ^Δ^ This is equivalent to 2000 mg/kg choline.

**Table 2 animals-12-01313-t002:** Effect of the polyherbal level and choline chloride (mg/kg) on the blood chemistry of dogs.

	Polyherbal				Choline Chloride ^Δ^		*p*-Value		
Item	0	200	400	800	3850	SEM	L	Q	K *
Glucose, mg/dL	65.90	57.57	55.08	56.49	58.61	4.346	0.12	0.27	0.65
Cholesterol, mg/dL	192.31	144.92	160.11	175.68	145.08	13.201	0.56	0.02	0.32
Urea, mg/dL	32.99	23.96	53.69	28.72	32.89	11.89	0.60	0.41	0.98
Uric acid, mg/dL	0.289	0.302	0.450	0.343	0.238	0.077	0.37	0.44	0.16
Creatinine, mg/dL	0.662	0.674	0.724	0.663	0.737	0.056	0.83	0.51	0.45
Total protein, g/dL	6.50	6.17	6.32	6.39	6.06	0.246	0.88	0.42	0.41
Albumin, g/dL	3.36	3.24	3.21	3.14	3.26	0.116	0.19	0.81	0.63
Globulin, g/dL	3.08	2.88	3.11	3.24	2.68	0.193	0.42	0.39	0.08
Bilirubin, mg/dL	0.474	0.595	0.577	0.671	0.574	0.088	0.15	0.88	0.70
ALP, UI/L	56.37	43.00	54.16	44.94	44.88	10.723	0.63	0.84	0.84
LDH, UI/L	529.4	490.9	394.2	494.0	449.3	67.8	0.50	0.31	0.89
AST, UI/L	51.78	42.24	52.72	53.69	51.08	6.772	0.59	0.44	0.84
Calcium, mg/dL	8.68	8.23	7.77	8.05	8.38	0.348	0.13	0.30	0.37
P, mg/dL	4.24	3.75	4.32	4.42	4.21	0.396	0.53	0.47	0.92

* Contrast: polyherbal vs. choline chloride; AST: aspartate aminotransferase; ALP: alkaline phosphatase; LDH: lactate dehydrogenase; P: phosphorus; SEM: Standard error of the mean; L: linear effect; Q: Quadratic effect. ^Δ^ This is equivalent to 2000 mg/kg choline.

**Table 3 animals-12-01313-t003:** Effect of the polyherbal level and choline chloride supplementation (mg/kg) on the blood count of dogs.

	Polyherbal				Choline Chloride ^Δ^		*p*-Value		
Item	0	200	400	800	3850	SEM	L	Q	K *
Hemoglobin, g/dL	16.69	17.16	16.89	16.69	16.75	0.459	0.88	0.44	0.76
Hematocrit, %	49.76	50.98	51.38	50.29	50.37	1.572	0.78	0.46	0.78
Erythrocytes, 10^6^/mL	7.45	7.75	7.46	7.37	7.53	0.267	0.66	0.46	0.98
Leukocytes, 10^3^/mL	18.02	12.18	11.51	14.52	11.15	1.644	0.90	0.14	0.40
Lymphocytes, %	19.95	27.84	23.42	24.81	26.32	3.032	0.46	0.28	0.77
Monocytes, %	2.68	2.54	2.91	3.37	2.80	0.813	0.51	0.71	0.88
Neutrophils, %	74.08	64.49	70.88	68.58	69.30	3.270	0.49	0.27	0.72
Eosinophils, %	3.22	5.10	2.73	3.30	2.23	0.702	0.50	0.35	0.07

* Contrast: polyherbal vs. choline chloride; L: linear effect; Q: quadratic effect; SEM = standard error of the mean. ^Δ^ This is equivalent to 2000 mg/kg choline.

**Table 4 animals-12-01313-t004:** Biological processes enriched by polyherbal when compared to choline chloride ^Δ^.

Potential Implications in Dog ^1^	Metabolic Pathway ^2^	Fc ^3^	*p*-Value
Prevention of cardiovascular and metabolic diseases	Renin–angiotensin system ^−^	3.4	0.03
	Carbohydrate absorption and digestion ^−^	3.3	0.02
	PPAR signaling pathways ^−^	2.5	0.02
Prevention of cancer	Signaling pathways that regulate stem cell pluripotency (SPRSCP) ^−^	2.4	0.001
	TGFβ signaling pathways ^−^	2.2	0.03
	Transcriptional dysregulation in cancer ^−^	2.3	0.002
	Estrogen signaling pathways ^−^	2.2	0.03
	Cholinergic synapse ^−^	2.1	0.03
	FoxO signaling pathways ^−^	2.1	0.02
	MAPK signaling pathways ^+^	2.0	0.01
	Pathways in cancer ^−^	1.5	0.04
Inflammatory and immune response	Adherens junctions ^−^	2.5	0.03
	Cytokine–cytokine receptor interaction ^−^	1.7	0.02
	Chemokine signaling pathways ^−^	1.7	0.04
Behavior and cognitive process	Addiction to nicotine ^−^	3.5	0.01

^1^ Authors classification; ^2^ analyses from DAVID 6.8 Bioinformatic gene set enrichment. ^3^ Fc: fold enrichment. PPAR: peroxisome proliferator associated receptor. TGFβ: transformer growth factor β. FoxO: forkhead box transcription factor. MAPK: mitogen-activated protein kinase. ^–^ Biological processes were enrichment with down-regulated genes. ^+^ Biological processes were enrichment with up-regulated genes. ^Δ^ This is equivalent to 2000 mg/kg choline.

**Table 5 animals-12-01313-t005:** Changes in gene expression in dogs fed the polyherbal vs. choline chloride in processes related to disease prevention and metabolism.

**Renin–Angiotensin System**	**Fc**
Angiotensinogen (serpin peptidase inhibitor, clade A, member 8) (*AGT*)	−2.1
Carboxypeptidase A3, mast cell (*CPA3*)	−1.6
Cathepsin A (*CTSA*)	−2.0
Chymase 1, mast cell (*CMA1*)	−2.0
Neurolysin (metallopeptidase M3 family) (*NLN*)	−1.6
Renin 1 structural (*REN1*)	−1.6
**Carbohydrate Absorption and Digestion**	**Fc**
Amylase 1, salivary (*AMY1*)	−1.7
Calcium channel, voltage-dependent, L type, alpha 1D subunit (*CACNA1D*)	−1.6
Glucose-6-phosphatase, catalytic (*G6PC*)	−1.7
Glucose-6-phosphatase, catalytic, 2 (*G6PC2*)	−1.7
Solute carrier family 2 (facilitated glucose transporter), member 2 (*SLC2A2*)	−1.6
Solute carrier family 37 (glucose-6-phosphate transporter), member 4 (*SLC37A4*)	−2.4
Thymoma viral proto-oncogene 1 (*AKT1*)	−2.0
**PPAR Signaling Pathways**	**Fc**
Aquaporin 7 (*AQP7*)	−2.3
Carnitine palmitoyltransferase 2 (*CPT2*)	−2.3
Cytochrome P450, family 4, subfamily a, polypeptide 10 (*CYP4A10*)	−1.8
Cytochrome P450, family 8, subfamily b, polypeptide 1 (*CYP8B1*)	−1.8
Matrix metallopeptidase 1a (interstitial collagenase) (*MMP1A*)	−1.7
Matrix metallopeptidase 1b (interstitial collagenase) (*MMP1B*)	−1.8
Phospholipid transfer protein (*PLTP*)	−1.6
Solute carrier family 27 (fatty acid transporter), member 2 (*SLC27A2*)	−1.7
Solute carrier family 27 (fatty acid transporter), member 5 (*SLC27A5*)	−1.6
Sterol carrier protein 2, liver (*SCP2*)	−1.7
Fatty acid-binding protein 7, brain (*FABP7*)	2.2
Cytochrome P450, family 27, subfamily a, polypeptide 1 (*CYP27A1*)	2.6
Peroxisome proliferator-activated receptor alpha (*PPARA*)	3.1

Fc: fold change.

**Table 6 animals-12-01313-t006:** Changes in gene expression of dogs fed with polyherbal vs. choline chloride in biological processes related to cancer.

SPRSCP		TGFβ		Estrogen		FoxO		Cancer *		MAPK *		TDC		Cholinergic Synapse	
Gene	Fc	Gene	Fc	Gene	Fc	Gene	Fc	Gene	Fc	Gene	Fc	Gene	Fc	Gene	Fc
*BMI1*	−1.7	*SMAD2*	−2.1	*FKBP4*	−1.8	*BCL6*	−3.3	*MECOM*	−1.9	*MAPKAPK2*	4.1	*BCL2A1B*	−1.7	*FYN*	−1.6
*HNF1A*	−2.3	*SMAD7*	−3.9	*ATF4*	−1.6	*SMAD2*	−2.1	*PTK2*	−1.9	*MKNK2*	1.9	*BCL6*	−3.3	*ATF4*	−1.6
*SETDB1*	−1.7	*AMH*	−2.2	*ADCY8*	−1.8	*FOXG1*	−2.0	*SMAD2*	−2.1	*TAOK2*	2.4	*BMI1*	−1.7	*ADCY8*	−1.8
*SMAD2*	−2.1	*BMP2*	−2.1	*HSPA2*	−1.6	*G6PC*	−1.7	*APC*	−1.7	*RELB*	2.3	*CD14*	−1.6	*CACNA1D*	−1.6
*APC*	−1.7	*BMP8B*	−1.8	*HSP90AB1*	−1.7	*G6PC2*	−1.7	*ADCY8*	−1.8	*FGF7*	2.2	*FLI1*	−1.9	*CAMK4*	−2.0
*AXIN2*	−1.8	*INHBA*	−1.8	*MMP9*	−1.6	*HOMER3*	−1.7	*AXIN2*	−1.8	*JUN*	1.8	*PTK2*	−1.9	*CHAT*	−1.6
*BMP2*	−2.1	*INHBC*	−1.6	*NRAS*	−1.7	*NRAS*	−1.7	*BMP2*	−2.1	*MAP2K6*	3.0	*WHSC1*	−1.9	*CHRNA7*	−1.8
*ESRRB*	−2.6	*IFNG*	−2.8	*PLCB4*	−1.7	*STK4*	−1.8	*CASP3*	−1.8	*MAP3K12*	2.2	*ELANE*	−1.6	*NRAS*	−1.7
*FGFR4*	−1.9	*PITX2*	−2.5	*KCNJ3*	−1.8	*SGK2*	−1.7	*CTNNA2*	−1.9	*MRAS*	1.7	*ETV6*	−1.7	*PLCB4*	−1.7
*HESX1*	−2.4	*TGFBR2*	−1.8	*SOS1*	−1.6	*SOS1*	−1.6	*CXCL12*	−1.7	*NTRK2*	2.0	*HMGA2*	−2.0	*KCNJ3*	−1.8
*INHBA*	−1.8	*CHRD*	−1.9	*AKT1*	−2.0	*SOD2*	−1.6	*FGF10*	−1.6	*PLA2G4E*	1.9	*HIST1H3G*	−2.1	*KCNJ4*	−1.9
*INHBC*	−1.6	*ID2*	−1.7			*AKT1*	−2.0	*FGF21*	−1.5	*PDGFRA*	1.9	*IL3*	−2.1	*AKT1*	−2.0
*NRAS*	−1.7	*ID4*	−1.5			*TGFBR2*	−1.8	*HSP90AB1*	−1.7	*PDGFRB*	2.8	*KMT2A*	−1.7		
*AKT1*	−2.0	*PPP2R1B*	−1.7			*USP7*	−1.9	*LAMA4*	−2.2	*PPP5C*	1.6	*MMP3*	−2.3		
*WNT1*	−1.6	*TGFBR1*	−2.1					*LPAR1*	−1.8	*PTPN5*	1.6	*MMP9*	−1.6		
*WNT7B*	−2.3							*MMP1A*	−1.7	*SOS2*	1.6	*REL*	−2.3		
*ZIC3*	−1.9							*MMP1B*	−1.8	*TGFBR1*	2.1	*RUNX1T1*	−1.9		
								*MMP9*	−1.6			*TGFBR2*	−1.8		
								*NRAS*	−1.7			*UTY*	−1.9		
								*PLCB4*	−1.7						
								*RARB*	−1.8						
								*RUNX1T1*	−1.9						
								*STK4*	−1.8						
								*SMO*	−1.8						
								*SOS1*	−1.6						
								*AKT1*	−2.0						
								*TGFBR2*	−1.8						
								*WNT1*	−1.6						
								*WNT7B*	−2.3						

* Signaling pathways; Fc: fold change; SPRSCP: signaling pathways that regulate stem cell pluripotency; TGF_β_: transforming growth factor β; Fox O: forkhead box O; MAPK: Mitogen-activated protein kinase: TDC: Transcriptional dysregulation in cancer.

**Table 7 animals-12-01313-t007:** Changes in the gene expression of dogs fed with the polyherbal vs. choline chloride in the inflammatory process and immune response.

Adherens Junctions		Chemokine Signaling Pathways		Cytokine–Cytokine Receptor Interaction	
Gene	Fc	Gene	Fc	Gene	Fc
*FYN*	−1.6	*GRK4*	−2.1	*AMH*	−2.2
*SMAD2*	−2.1	*PTK2*	−1.9	*BMP2*	−2.1
*ACTG1*	−2.0	*ADCY8*	−1.8	*XCR1*	−2.2
*AFDN*	−2.4	*XCR1*	−2.2	*CCL12*	−1.6
*CTNNA2*	−1.9	*CCL12*	−1.6	*CCL6*	−1.6
*CTNND1*	−1.7	*CCL6*	−1.6	*CCR1L1*	−1.9
*NECTIN2*	−2.2	*CCR1L1*	−1.9	*CCR7*	−1.8
*PTPRJ*	−1.7	*CCR7*	−1.8	*CXCL11*	−2.4
*TGFBR2*	−1.8	*CXCL11*	−2.4	*CXCL12*	−1.7
		*CXCL12*	−1.7	*CXCL9*	−1.7
		*CXCL9*	−1.7	*CSF2RB2*	−1.5
		*NRAS*	−1.7	*EPOR*	−1.7
		*PLCB4*	−1.7	*INHBA*	−1.8
		*PPBP*	−2.0	*INHBC*	−1.6
		*STAT2*	−1.6	*IFNG*	−2.8
		*SOS1*	−1.6	*IL3*	−2.1
		*AKT1*	−2.0	*PPBP*	−2.0
				*TGFBR2*	−1.8
				*TNFSF9*	−2.2
				*TNFRSF11B*	−1.6
				*TNFRSF1A*	−2.5

Fc: Fold change.

**Table 8 animals-12-01313-t008:** Changes in gene expression in dogs fed with polyherbal vs. choline chloride in nicotine addiction processes.

Gene	Fc
Cholinergic receptor, nicotinic, alpha polypeptide 7 (*CHRNA7*)	−1.8
GABA A receptor, subunit alpha 1 (*GABRA1*)	−2.3
GABA A receptor, subunit alpha 4 (*GABRA4*)	−1.8
GABA A receptor, subunit beta 2 (*GABRB2*)	−1.8
GABA A receptor, subunit theta (*GABRQ*)	−2.0
Solute carrier family 17 (sodium-dependent inorganic phosphate cotransporter) (*SLC17A7*)	−2.3
Solute carrier family 32 (GABA vesicular transporter), member 1(*SLC32A1*)	−2.4

Fc: fold change; GABA: Gamma-aminobutyric acid.

## Data Availability

The data presented in this study are available in the insert article or Appendix A here.

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
