# Peer review of "Influence of a Polyherbal Choline Source in Dogs: Body Weight Changes, Blood Metabolites, and Gene Expression"

_animals, 2022, doi:10.3390/ani12101313_

Round 1

Reviewer 1 Report

General comments

There is the need of minor touches on the formatting of some tables.

Introduction – you mentioned in the abstract that choline chloride is hygroscopic and a challenge to work with in manufacturing settings. Why not explore this a little further in the introduction as well? I think this is characteristic of choline chloride helps you to create a even stronger case to replace it by the herbal mix.

Discussion – overall you did a great job to explain possible mechanisms of several different genes that were affected by the use of the herbal mix; however, it is not clear if the references used are based on research with dogs or not. If not, do you know if these mechanisms would also happen in the dog? In this case, I think you must make clear that this is a possibility, but not a certainty. One thing that would be beneficial is to have an overall impact of the herbal mix at the end of each session in the discussion considering the effects of all the affected genes combines, what do the authors expect that it would happen? This could be a good summary within each sub-session for the readers.

Specific comments

L23 “In pet foods” not “In pet feeds”

L33 “choline in dog foods” not “choline in food dogs”. And use semicolon before “however” and a coma after

L37 “housed” what? Individually? In groups?

L38 replace “next” by “following”

L41-42 daily weight gain was not significant for you to report it as such. Please revise this sentence

L46-47 I think it is important here you state that this was a relatively short feeding period. I would not fully trust a replacement unless a long-term feeding trials are performed.

L53-54 500mg and 2000mg per kg of diet? Per kg of body weight?

L55-56 400mg/kg of food? body weight?

L61 what are the references for this statement? What other species?

L63-65 I don’t think the need to reevaluate the choline requirements is related to the work you did here. I would remove this sentence from here.

L74 “improves antioxidant status at the cellular level”

L76-77 “liver weight” or live weight? On table 1 you report body weight, not liver weight

L89-90 I don’t think this information is relevant for the trial. If the dogs were not kept in controlled environmental conditions, then you would need to specify the humidity and temperature (average and min-max range) as well as light-dark cycles.

L111-113 I think there is something missing here between “water” and “adjusting” please revise

2.1. – In this session it would be important to describe for how long the animals were feed the different diets.

2.2 – were the dogs evaluated prior to the study to know they had no issued before starting the trial? They could have some pre-existing conditions that would last throughout the trial and these conditions could be misinterpreted as results of the treatments.

L124 in the abstract you mention that blood collection was done on day 59. What is the correct day of collection?

L126-127 what was the private lab name?

L138 “immediately” what?

L140 how the samples were homogenized?

L170-175 if they were not significant, why are you giving importance to it by reporting the numerical differences? You should only report that there was not a treatment effect on ADG.

L176-177 your dorsal fat contrast C is not significant as what you outlined on session 2.5. P<0.05 is different than P≤0.05. Please revise.

Table 1 – ADG not defined. I think the unit for the Energy intake should be kcal/d. Could you explain why you think it is Mcal/d?

L180-181 You should ass to session 2.5 what is the p-value interval that you considered a trend. “…glucose as herbal mix concentration increased in the diet.”

L184 P=0.08 is not a difference, it is a tendency, please revise this sentence

L199-204 maybe it would be beneficial to have these 2207 genes affected by the herbal mix outlined on a supplemental file along with the magnitude of the change (2x, 4x, 0.5x). This could be used as a reference if other authors perform research with other choline sources

Table 4 PPAR, TGF, FoxO, MAPK not described. It would be interesting to add a plus or minus sign in front of the fold change to indicate if the change was an increase or decrease

Table S1 – I think it would be beneficial to have this table added to the main manuscript rather than as supplemental material

Table 7 – you could use the abbreviation GABA and add a footnote explaining what it means to facilitate reading this table.

L239-241 I don’t think that is the case, because we don’t know for how long the dogs were fed these diets. I think it is premature to make such a strong statement about the efficacy of this herbal mix.

L262 I think you are missing a period after “liver”, instead of the semicolon

L276 “choline supplemented dairy cattle”

L277 showed and not “show”

L284-286 why there were no issues with this lower dietary choline level? This is not clear from the data you present. I think this could be an issue related to the length of the experiment, it could have been too short of a trial to evaluate the impact of long-term feeding this herbal mix or deficient diets.

L292 “retention of sodium and water” or please revise this sentence

L330 I think you are missing a period after “studies”

L341 “Andrographis paniculata” needs to be italicized

L375 FoxO instead of FOXO?

L443-444 I think in this line you need to make clear that the study was a “short” study by indicating the duration of the trial. This could be a big difference from the results you report here and a long-term study. It is good that you acknowledge that long-term trials are needed in the next sentence, but I am not completely convinced that you could replace choline by the herbal mix. Also, you MUST make it clear that this is for adult dogs, not puppies and not gestating dogs. These other life stages could have different needs than adult dogs. One conclusion that you could explore is the safety for the animal. This study indicated that this ingredient is not toxic for adult dogs, which is a good indication that it would be safe to feed to other life stages as well.

Reviewer 2 Report

General comments

This is a novel and important study that evaluated the effects of the replacement of choline chloride by an herbal source on body weight, blood variables, liver ultrasound, and gene expression in dogs. I recommend the study for publication after major revision, considering the following:

  • The English needs to be reviewed. There are a lot of English mistakes. I pointed out only some of them in the specific comments.
  • The authors need to standardize the name of the experimental treatments. E.g.: sometimes they call herbal choline, and sometimes polyherbal; synthetic choline or choline chloride.
  • Some important words are missing to improve the sentences, and this also highlights the importance of grammar checking. E.g.: L43: “reduced glucose and cholesterol levels” is missing “blood” or “serum” glucose and cholesterol levels. “Dogs with choline sources” is strange, it may be changed to: dogs fed diets with choline sources.
  • They need to clarify some important statistical issues. E.g. they can`t say that some result were a trend with a higher P-value (higher than 0.10) and without defining it in the statistical analysis section. Also, they can`t discuss results that were not statistically different.

Specific comments:

Title

Change to: … “body weight” instead of “live weight”. Change live weight to body weight throughout the text.

Change to: …polyherbal choline source…

Simple summary

L29: Change to: supply of “choline”, instead of “vitamin” - (choline is not a true vitamin)

L37: housed where?

L41: for “the” two choline sources

L42: … that “received” the polyherbal “diet”…

L41-42: when you present a trend is better to indicate the exact P-value (instead of P<0.10).

Introduction

L51: Remove the word “organic”.

L52: dog food instead of food dogs

L56: …without affecting…

L62: …companion animals… instead of company…

Materials and methods

I think is better to standardize all the diets to the inclusion level of the choline sources and then cite all the choline concentrations of the treatments (It is not clear how much of choline chloride was added in the diet). Those choline concentrations were analyzed (and by which method) or calculated?

L99: The basal diet was a complete dry extruded dog food for maintenance… that was formulated… 

Include the body condition score of the dogs.

L156: include the contrast that compared the choline chloride vs. herbal choline. Include the number of replicates/treatment and the experimental design. Also, you need to specify which P-values are considered trend (usually between 0.05 and 0.10).

Which statistical analysis was used to evaluate the gene enrichment differences?

Results

L170-172: If there were no statistical differences you can`t say that losses were observed. You also can`t say that some treatments tended with a P-value of 0.20 or 0.12 (L180).

L177: If you run a correlation analysis you need to cite and explain in materials and methods.

Table 1 and other tables

  • Why 800 mg/kg is not bolded?
  • You should change to Polyherbal and Chloride instead of Choline (and put the inclusion level of the chloride instead of choline) because is strange to name some treatments by the inclusion level of the additive and some by the dietary choline concentration.
  • What is ADG? You need to inform the readers in the footnote.

L195-196: Comment if the liver results were not shown or if they are in supplementary data.

Discussion

Be careful, you can`t discuss the variables that were not statistical difference among treatments or that the P was higher than 0.10 (can not be considered even a trend). Same comment for the conclusion.

Reviewer 3 Report

This study investigated the use of biocholine from a polyherbal mix on blood metabolites and different gene expressions in blood using microarray. The study is useful and has novelty.

I have few comments

L93-95: the breed distribution among the treatments are not clear. 

Table 1: why was SEM value for ADG so high compared with the mean values?

For 800 mg, ADG should be positive?

Use unit of BW.

All tables: C may be changed as it looks like cubic effect.

Round 2

Reviewer 2 Report

The authors made improvements, but there are some issues that need to be considered. The English must be reviewed, there are some sentences that do not make sense, e.g.: L64-65: “If in broilers, and ruminants [2-5] has been possible to meet choline requirements with PtdCho and it is necessary to evaluate different alternatives to provide this nutrient in companion animals.” Other specific comments are:

L48: remove: “…improving glucose…” You can`t affirm that.

L179-180: review English.

Table 1: replace Feed intake g/d in the table by FI.

L185: Replace the sentence by: Dogs fed diets… “presented” lower…

L186: Remove the sentence about blood glucose because it did not differ among treatments.

Table 7: Review and correct the title.

L434-436: Review this sentence.
